

# Screen time exposure and academic performance, anxiety, and behavioral problems among school children

Mohammad Sidiq[1,2,*], Balamurugan Janakiraman[3], Faizan Kashoo[4], Rayan Jastania[5], Abdullah Ibrahim Alhusayni[6], Abdullah Alzahrani[6], Aksh Chahal[1,2], Alagappan Thiyagarajan[7], Imran Khan[8], Chandan Kumar[1], Rajkumar Krishnan Vasanthi[9], Fahad Alanazi[10], Mehrunnisha Ahmad[11], Chhavi Arora Sehgal[12], Shabnam Khan[12] and Mshari Alghadier[13,*]

[1] Department of Physiotherapy, School of Allied Health Sciences, Galgotias University, Greater Noida, Uttar Pradesh, India
[2] Galgotias Multi-Disciplinary Research and Development Cell (G-MRDC), Galgotias University, Greater Noida, UP, India
[3] SRM College of Physiotherapy, Faculty of Medicine and Health Sciences, SRM Institute of Science and Technology (SRMIST), Chennai, Tamil Nadu, India
[4] Department of Physical Therapy and Health Rehabilitation, College of Applied Medical Sciences, Majmaah University, Al Majmaah, Saudi Arabia
[5] Department of Physical Therapy, Faculty of Medical Rehabilitation Sciences, King Abdulaziz University, Jeddah, Saudi Arabia
[6] Department of Health Rehabilitation, College of Applied Medical Sciences, Shaqra University, Shaqra, Saudi Arabia
[7] Department of Physiotherapy, Chettinad Academy of Research and Education, Chennai, India
[8] Department of Physiotherapy, University of Engineering and Management, Jaipur, India
[9] Faculty of Health and Life Sciences, INTI International University, Nilai, Malaysia
[10] Department of Physical Therapy and Health Rehabilitation, College of Applied Medical Sciences, Jouf University, Aljouf, Saudi Arabia
[11] College of Nursing, Majmaah University, Riyadh, Saudi Arabia
[12] Centre for Physiotherapy and Rehabilitation Sciences, Jamia Millia Islamia University, New Delhi, India
[13] Department of Health and Rehabilitation Sciences, College of Applied Medical Sciences, Prince Sattam Bin Abdulaziz University, Riyadh, Saudi Arabia
* These authors contributed equally to this work.

Corresponding author
Mshari Alghadier,
m.alghadier@psau.edu.sa

## ABSTRACT

**Background:** The growth of the digital landscape has surely outpaced research on the effects of screen media on the health, learning, and development of children. The potential risk-to-benefit ratio of screen media exposure for education and entertainment purposes warrants further exploration. Therefore, we aimed to investigate the relationships between screen time and academic performance, anxiety, and outdoor playing among school children in India.

**Methods:** A total of 537 parents responded to this online survey and reported approximately 537 Indian school children (mean age 10.9 years) from five schools. Data was collected using an e-questionnaire which consisted of a socio-demographic domain, recreational activities, self-reported physical measures, academic performance, information related to children's screen time, the Spence Children Anxiety Scale (SCAS), and the Pediatric Symptom Checklist (parent version) instruments. We employed multivariate linear regression models to estimate the

association between children's screen time and the predictor variables with 0.05 alpha as level of significance.

**Results:** The mean screen time was 3.06 ± 1.22 h/day, the average duration of outdoor play per week was 11.23 ± 4.1 h, and the mean body mass index (BMI) was 18.2 ± 3.2. Screen time (h) in a typical week was positively correlated with BMI, the SCAS anxiety score, and behavioral problems and negatively correlated with academic performance. School children's screen time was a predictor of their BMI, behavioral symptoms, and academic performance according to the linear regression analysis.

**Conclusion:** Our findings pointed out that screen time was associated with increased BMI, behavioral problems and poor academic performance. These insights shall be used for development of targeted tailored interventions and strategies to reduce childhood obesity related to screen time. Further research is necessary to rule out the influence of other intricate factors, such as sleep, parental practices, family connectedness, and supervision of parents. The careful use of digital media must serve as a positive force in children's educational and developmental trajectories.

# BACKGROUND

Digital learning is already an essential part of Indian schools. The Digital India Education Program (Prime Minister E-Vidya) aims to improve the digital infrastructure by providing school children with digital content (*Kumar & Selva Ganesh, 2022*; *Asif & Panakaje, 2022*). The National Council of Education Research and Training (NCERT, India) offers school children interactive learning, textbooks, audio, and video content (*Singh & Agarwal, 2023*). Although few rural areas in India lack affordable internet and digital infrastructure, the digital divide persists. The Union Education Ministry (UDISE, 2021–22) reported that 0.54 million of the 1.48 million schools used digital learning systems (*Deb, 2024*). Digital education is a government initiative and the future for many Asian countries. Digital education offers interactive learning, local language content, different formats, customized classes, inclusivity, and flexibility (*Li et al., 2020*). The health concerns of children must be explored further if we are to safely transition to technology-based education. According to Happinetz, Indian children spend 2–4 h per day on screens for entertainment and digital games. Screen time exposure harms children's physical, mental, social, and academic health, so researchers recommend limiting it (*Moitra & Madan, 2022*; *Varadarajan et al., 2021*). Inadequate eating practices also seem to be linked to the association between screen time and body mass index (BMI); this is a phenomena that has been seen in Mediterranean cultures, where more screen exposure is linked to decreased intake of fruits and vegetables (*Wärnberg et al., 2021*).

Screen time exposure is reported as early as infancy in most countries, and the prevalence of excess screen time ranges from 10% to 93.7% across high-income countries,
21% to 98% in middle-income countries, and 1.0 to 3.1 h/day among school-aged children in several countries (*Carson & Janssen, 2012*; *Domingues-Montanari, 2017*; *Hale & Guan, 2015*; *Oswald et al., 2020*). Studies conducted in India reported that the median age of first screen exposure was 10 months, and most school-aged children used smartphones (96%), followed by television (89%). A systematic review of Indian studies revealed that 39% to 44% of adolescents suffer from smartphone addiction, the most likely root cause of which is excess screen time during childhood. Studies report that screen time of >2 h/day is associated with myopia risks, depression, obesity, and sleep disorders among children.

The Indian Academy of Pediatrics (IAP) advises that children aged 5 years or younger should have less than 1 h of screen time, whereas older children should balance screen time with physical activity, sleep, meals, academic responsibilities, social interactions, and hobbies. If screen time appears to supplant these activities, it can be classified as excessive screen (*Gupta et al., 2022*; *Sharma et al., 2022*). In addition, there is significant variation in the definition of excess screen time across studies, ranging from 1 h to more than 3 h (*Muppalla et al., 2023*; *Pandya & Lodha, 2021*; *Tezol et al., 2022*). In 2021, the Indian Academy of Pediatrics (IAP) released parental guidelines regarding screen time, cautioning against the detrimental effects of excessive usage and providing recommendations on digital hygiene, appropriate usage, and the initiation of social media for children and related subjects (*Sharma et al., 2022*).

The recent surge in digital usage and associated health concerns such as sleep disturbances, shortened sleep duration, adverse weight gain, poor academic performance, attention deficits, emotional dysregulation, less prosocial behavior, reduced physical activity, and outdoor play among children has drawn the attention of researchers (*Menon, Mishra & Padhy, 2021*; *Presta et al., 2024*). Conversely, some researchers argue that screen time is associated with better academic performance. We hypothesized that there is a linear relationship between screen time among children with their nutritional status, behavior, and academic performance. Hence, given the governmental plan to expand digital technology in Indian schools and the reports of digital gluing among children, this study aimed to explore the relationships between screen time and academic performance, anxiety, and outdoor playing among school children in India.

## METHODS

### Study design

A quantitative web-based cross-sectional design was used from January 2024 to April 2024 among parents of school children in Greater Noida city, India. The study was approved by the Departmental Research Committee, School of Allied Health Sciences, Galgotias University, under reference to DEC/FEA/PT/06/23, and the study was registered with the Indian Council of Medical Research India (CTRI/2023/10/058965). This study adhered to the General Data Protection Regulation (GDPR) and the Digital Personal Data Protection Act, India (*Prasad & Menon, 2020*).

## Participants and consent

The school authorities of private schools in Greater Noida city were approached, and the importance of the study was explained. From the 21 private schools approached, the counselors of five private schools consented for the study, and the counselors distributed the online survey questionnaire structured by the investigators to the parents of children in grades 3 to 9. Considering the higher non-response rate reported by studies in the past, we decided to distribute the survey questionnaire to all the parents. Hence, no assumptions were made for calculation of power sample.

The parents confirmed their participation by providing informed consent at the beginning of the online survey. The consent form and information sheet emphasized the right to withdraw from the study without consequences. The anonymized dataset was obtained from school counselors.

## Study area

The five private schools included in this study followed the Central Board of Secondary Education (CBSE) curricula. All the included school have facilities like spacious classrooms, playgrounds, and sports activity. The schools adhered to the fee structure of the National Education policy, India (NEP) and most of the parents represent the middle and high socio-economic strata.

## Data collection and survey instrument

We designed a self-administered online questionnaire with 95 items, which was compatible with mobile devices, and tablets employing a user-friendly interface. To facilitate easy response, we used brief instructions, simple choices, big fonts, and mostly select options. Different links were created for each grade of study to allow multiple responses if the parent had more than one child studying in the five schools selected. Yet, there are potential limitations like self-selection bias, poor response, response quality, and internet access. In addition, parental reporting over self-reporting by children might induce surrogate bias.

The questionnaire gathered information regarding the demographic profile of the child. Anthropometric measures, such as the height and weight of the children, as well as the average screen time in hours per day and the average academic performance score of the current year were recorded. Screen-related questions such as screen media availability, frequency of use by the child, and use time were used. The other measures used are described below.

The Spence Children Anxiety Scale (SCAS-Child): This scale consists of 45 items, 38 of which assess specific anxiety symptoms, and the remaining seven are filters used to reduce response bias. It is rated on a four-point Likert scale where 0 is "never", 1 is "sometimes", 2 is "often" and 3 is "always". The total score is calculated from six subscales (panic/agoraphobia, separation anxiety, social phobia, obsessive compulsive disorder and physical injury fears). The total score ranges from 0–114. A higher score indicates greater severity. The average for each subscale is calculated to estimate the severity of

anxiety-related symptoms (*Glod et al., 2017*). The SCAS is a valid tool (ranging from 0.61–0.84) and reliable (ranging from 0.71–0.90) (*Toscano et al., 2020*).

Pediatric Symptom Checklist (PSC-Parent version): this scale was used to identify emotional and behavioral problems (*Muzzolon, Cat & dos Santos, 2013*). It consists of 35 items where parents rate a child's behavior as "never", "sometimes" or "often". The total score is calculated from 0 to 20, where a higher score indicates a greater likelihood of emotional or behavioral problems. The school counselors were requested to send reminders to the parents to complete the questionnaire after 7 days, and 537 responses were collected after 15 days.

The SCAS-Child and PCS-Parent version tools were reported to be used in the Indian context extensively and found to be suitable (*Karande et al., 2018*; *Russell et al., 2013*; *Thakkar et al., 2016*). The study is reported in accordance with the Strengthening the Reporting of Observational studies in Epidemiology (STROBE) recommendations (*Elm et al., 2007*).

### *Post hoc*-power analysis

As the feasible sample size was not known, no priori analysis was conducted. Using the G*power software (V 3.1.9.6) for Windows, a *post-hoc* power analysis was performed to determine the achieved power for multiple linear regression in this study. The study's actual sample size is 537, the fixed models, $R^2$ deviation from '0', a detectable effect size of F squared 0.1 (small effect), an alpha of 0.05, and number of predictors ($n = 3$). The calculated *post-hoc* achieved power ($1 - \beta$) using the aforementioned inputs was 0.97, which exceeds the recommended threshold beta value of 0.80. Therefore, there is enough power to back up the findings of this investigation.

### Data analysis

The data were analyzed *via* SPSS version 26.0 for Windows (IBM Corp., Armonk, NY, USA). The assumption of a normal distribution of the continuous variables was tested *via* the Kolmogorov–Smirnov test, and the data are presented as the means ± standard deviations (SDs). The categorical variables are presented as frequencies (percentages). Descriptive statistics were used to describe the children's demographic characteristics, parent-reported screen time, play time, academic performance, pediatric symptom score, and child anxiety score. Pearson correlation analysis was used to assess the direct correlation between screen time and other variables, with the level of significance at <0.05. Linear regression analysis was used to explore whether the parent-reported screen time of school children predicts variables such as BMI, play time, academic performance, SCAS-Child score and PSC-Parent score. When a clear sub-group existed, analyses were performed to determine the main effects and interaction effects of independent variables over the dependent variables. There were no missing data in this study.

## RESULTS

### Baseline characteristics

A total of 537 parents, comprising 291 boys (54.2%) and 246 girls (45.8%), responded to the e-questionnaire. The age of the children ranged from 8–14 years, with a mean age of

**Table 1** Basic characteristics of all participants.

| Variables | Descriptive statistics |
|---|---|
| Sex | |
| Boys (*n*, %) | 291, 54.2% |
| Girls (*n*, %) | 246, 45.8% |
| Age, mean (SD) | 10.98 ± 2.01 |
| BMI, mean (SD) | 18.1 ± 3.2 |
| Screen time (h)/week, mean (SD) | 21.45 ± 8.54 |
| Play time (h/week), mean (SD) | 11.23 ± 4.1 |
| Academic performance %, mean (SD) | 81.1 ± 9.6 |
| Pediatric symptom checklist mean (SD) | |
| Attention problem PSC | 5.04 ± 1.8 |
| Internalizing problem PSC | 5.1 ± 1.9 |
| Externalizing problems PSC | 7.2 ± 2.7 |
| PSC total | 38.4 ± 12.4 |
| Spencer's children anxiety scale mean (SD) | |
| Separation anxiety score | 6.1 ± 2.3 |
| Social phobia score | 6.2 ± 2.2 |
| Obsessive compulsive score | 6.3 ± 2.2 |
| Panic agoraphobia score | 9.1 ± 3.3 |
| Physical injury fears score | 5.1 ± 2.1 |
| Generalized anxiety score | 6.0 ± 2.4 |
| SCAS-Child total | 46.1 ± 14.7 |

**Note:**
Pediatric symptom checklist (PCS-Parent completed version): 74 maximum score from 37 items with three domains (Attention Problem PSC, Internalizing Problem PSC, Externalizing Problems PSC); Spencer's children anxiety scale: 135 is the maximum score from 45 items. Six domains of the scale were calculated separately (the separation anxiety score, social phobia score, obsessive compulsive score, panic agoraphobia score, physical injury fear score, and generalized anxiety score).

10.98 ± 2.01 years. The mean BMI of the school children, which was computed on the basis of the parents' reported height and weight, was 18.1 ± 3.2 kg/m$^2$, and academic performance, measured as the overall percentage of scores secured in the examination, was 81.13 (range 50–100%). The average play time in a typical week was 11.23 ± 4.1 h, and the total screen time across all digital displays was 21.45 ± 8.54 h per week (Table 1).

The average PSC (parent completed version) score for pediatric symptoms was 38.4 ± 12.4, with the subdomain of externalizing problems observed with a higher score (7.2 ± 2.7) and the average SCAS-Child version score was 46.1 ± 14.7, with the subdomain of panic agoraphobia observed with a higher score (9.1 ± 3.3).

## Correlation analysis

The correlations between screen time, age, BMI, play time, PSC score, and SCAS score are shown in Table 2. Screen time per week had no relationship with age or play time ($p > 0.05$). Screen time was positively correlated with children's BMI (r = 0.522, $p < 0.001$), academic performance (r = 0.371, $p < 0.001$), the PSC version of the pediatric symptoms

**Table 2  Correlation matrix of the sample ($n$ = 537).**

| | Age | BMI | PA | AcP | PSC | | | | SCAS-Child version | | | | | | |
| --- | --- | --- | --- | --- | --- | --- | --- | --- | --- | --- | --- | --- | --- | --- | --- |
| | | | | | AP | IP | EP | Total | SAS | SPS | OCS | PAS | PIFS | GAS | Total |
| Screen-time | 0.026 | 0.522** | 0.028 | -0.37** | 0.434** | 0.398** | 0.455** | 0.504** | 0.481** | 0.430** | 0.414** | 0.451** | 0.412** | 0.451** | 0.526** |
| Age | 1 | 0.01 | 0.035 | -0.021 | -0.049 | -0.071 | -0.014 | -0.041 | 0.001 | -0.025 | -0.051 | -0.009 | 0.041 | -0.011 | -0.018 |
| BMI | | 1 | 0.085 | -0.429** | 0.777** | 0.771** | 0.847** | 0.960** | 0.805** | 0.804** | 0.771** | 0.851** | 0.795** | 0.807** | 0.967** |
| PA | | | 1 | -0.057 | 0.067 | 0.048 | 0.082 | 0.093 | 0.096* | 0.073 | 0.060 | 0.071 | 0.051 | 0.061 | 0.081 |
| AcP | | | | 1 | -0.334** | -0.349** | -0.390** | -0.413** | -0.381** | -0.332** | -0.338** | -0.389** | -0.346** | -0.348** | -0.431** |
| PSC-AP | | | | | 1 | 0.593** | 0.637** | 0.814** | 0.634** | 0.599** | 0.626** | 0.610** | 0.643** | 0.611** | 0.743** |
| PSC-IP | | | | | | 1 | 0.645** | 0.737** | 0.572** | 0.621** | 0.740** | 0.697** | 0.594** | 0.687** | 0.790** |
| PSC-EP | | | | | | | 1 | 0.816** | 0.729** | 0.781** | 0.675** | 0.780** | 0.741** | 0.698** | 0.880** |
| PSC-total | | | | | | | | 1 | 0.779** | 0.770** | 0.736** | 0.812** | 0.750** | 0.781** | 0.928** |
| SCAS-SAS | | | | | | | | | 1 | 0.665** | 0.593** | 0.679** | 0.651** | 0.633** | 0.825** |
| SCAS-SPS | | | | | | | | | | 1 | 0.611** | 0.690** | 0.619** | 0.677** | 0.842** |
| SCAS-OCS | | | | | | | | | | | 1 | 0.623** | 0.616** | 0.626** | 0.793** |
| SCAS-PAS | | | | | | | | | | | | 1 | 0.681** | 0.682** | 0.878** |
| SCAS-PIFS | | | | | | | | | | | | | 1 | 0.627** | 0.818** |
| SCAS-GAS | | | | | | | | | | | | | | 1 | 0.842** |
| SCAS-Total | | | | | | | | | | | | | | | 1 |

**Notes:**
* $p < 0.05$.
** $p < 0.001$.
BMI, Body mass index; PA, play activity; AcP, academic performance; PSC, pediatric symptom checklist (PCS-parent completed version): with three domains (AP, attention problem PSC; IP, internalizing problem PSC; EP, externalizing problems PSC); Spencer's children anxiety scale: with six domains (SAS, separation anxiety score; SPS, social phobia score; OCS, obsessive compulsive score; PAS, panic agoraphobia score; PIFS, physical injury fear score; and GAS, generalized anxiety score).

**Table 3 Multiple linear regression analysis.**

| Model | Unstandardized coefficients β (SE) | Standardized coefficients β | t | Sig | 95% CI | r² |
|---|---|---|---|---|---|---|
| BMI | 0.989 (0.17) | 0.352 | 5.84 | <0.001 | [0.66–1.32] | 0.641 |
| Academic performance | −0.120 (0.034) | −0.134 | −2.96 | 0.003 | [−0.16 to −0.03] | |
| Pediatric symptom total score | 0.178 (0.04) | 0.259 | 4.33 | <0.001 | [0.09–0.26] | |

**Notes:**
BMI, Body mass index; β, beta coefficient; CI, confidence interval; r², coefficient of determination.

scale total score (r 0.504, $p < 0.001$) and subscales (attention problem (r = 0.434, $p < 0.001$), internalizing problems (r = 0.398, $p < 0.001$), and externalizing problems (r = 0.455, $p < 0.001$)). Screen time was also positively correlated with the SCAS total score for anxiety (r = 0.526, $p < 0.001$) and all its subscales. There was a wide range of positive correlations among the total scores and subscales of the PSC version of the pediatric symptoms scale and the SCAS scores (Table 2). No correlation was observed between age, play time and BMI ($p > 0.05$), whereas age and BMI demonstrated moderate positive correlations with the PSC version of the pediatric symptoms scale and the SCAS score, respectively.

### Linear regression analysis

The statistically significant variables in the correlation analysis were incorporated into the linear regression model (Table 3). The linear regression model indicated that screen time was associated with children's BMI (β − 0.98, t = 5.8, $p < 0.001$), academic performance (β − 0.12, t = −2.96, $p < 0.001$), and pediatric symptoms scale score (β − 0.178, t = 4.33, $p < 0.001$).

## DISCUSSION

This study assessed the associations between screen time and the academic performance, anxiety, and behavioral problems of school-aged children in India. The *post-hoc* power analysis suggests that the findings reported in this study have enough power to detect a small effect size. The key finding of this study is that children with excess screen time were observed to have poorer academic performance and higher levels of anxiety, alongside an increased body mass index (BMI). Modifiable factors, such as limiting screen time and prompting physical activity, are needed to support children's mental and physical health. Further, children's play time had no statistical association with screen time in this study unlike other reported studies which demonstrated higher sedentary behaviors with excess screen time (*Gupta et al., 2023*; *Ishtiaq et al., 2021*). The rationale might be the context of unstructured and structured sports, in addition to the multifactorial and multi-contextual nature of physical activity among children making it difficult to record. More importantly, play time and screen time may not be related sometimes, since this study also observed, children with recommended play time and with excess screen time.

### Amount of screen time

This study revealed that children are engaged in excessive screen time, as reported by parents, exceeding 3 h per day, among relatively young children aged 4–14 years. These
figures are three times greater than the recommended 1–2 h per day (*Saunders & Vallance, 2017*). There are studies reporting serious health issues with more than the recommended amount of screen time (*Dahlgren et al., 2021*; *Domingues-Montanari, 2017*; *Zhang et al., 2022*). The average screen time reported in this study is well beyond the WHO general recommended limit of 1 h a day. However, recreational or entertainment screen time and weekend screen time need to be considered separately to better understand the temporal pattern of screen time over the week (*Pardhan et al., 2022*).

## Cognitive and psychological implications

We found a higher score (35.7) of psychosocial distress among children assessed by the Pediatric Symptom Checklist (clinical cutoff of 28). The mean score of the internalizing problems domain of the scale assessing anxiety and depression (5.1) was marginally above the cutoff (5.0). A study conducted in the UK involving 14,665 participants revealed that adolescents who used computers for more than 3 h on weekdays had a 30% increased risk of anxiety and depression compared with those with less than 1 h of use (*Khouja et al., 2019*). In our study, the mean score of the externalizing problems domain of the scale assessing aggression and hyperactivity (7.2) was also marginally above the cutoff (7.0). In line with our findings, a systematic review and meta-analysis published in JAMA recently reported a small but significant association between greater screen time and both externalizing (*e.g.*, aggression, inattention) and internalizing (*e.g.*, anxiety, depression) behavior issues in children aged 12 years or younger (*Eirich et al., 2022*). In contrast, the mean score for attention problems was 5.05 (SD ± 1.9), below the cutoff of 7, indicating that these issues are less prominent. Contradictory evidence is available on the effect of screen time on attention. A systematic review suggested that attention decreases with increasing screen time. However, one study included in the same review suggested improvement in attention with increasing screen time among children (*Santos et al., 2022*). Another recently published study reported no relationship between screen time and attention subdomains among children aged between 6 and 10 years (*Liebherr et al., 2022*). The authors also suggest future research investigating the effect of socioeconomic status and a longitudinal study. Other studies have reported that sleep mediates the relationship between attention and screen time (*Guerrero et al., 2019*). Previous studies reported significant socio-economic status (SES) disparity regarding affordability, screen time, and physical activity opportunities. Since this study included children from private schools exclusively, the possibilities of SES disparity were not explored (*Aliyas, Mahmoudian & Cloutier, n.d.*).

## Body mass index (BMI), academic performance, screen time, and anxiety symptoms

The children in our study population fell into the normal weight category. However, an increase in BMI is strongly correlated with anxiety domains. The panic agoraphobia score (PAS) is a severe form of anxiety in certain situations. There was a strong correlation between increased BMI and panic agoraphobia score (r = 0.851), generalized anxiety score (GAS) (r = 0.807), separation anxiety score (SAS) (r = 0.805), social phobia score (SPS)

(r = 0.804), physical injury fear score (PIFS) (r = 0.795) and obsessive-compulsive score (OCS) (r = 0.771). These results suggest that children who engage in excessive screen time tend to have higher BMIs and reduced social interaction, which may lead to this severe form of anxiety. A review article reported the physical and psychological side effects of screen time and the development of anxiety (*Priftis & Panagiotakos, 2023*). Many other studies have reported the psychological effects of excessive screen time, ranging from sleep disturbance to severe psychological syndromes (*Khouja et al., 2017*; *Leung & Torres, 2021*; *Lissak, 2018*; *Mougharbel et al., 2023*; *Santiago et al., 2022*). Conversely, our findings strongly indicate that physical activity serves as a protective factor against the adverse effects of excessive screen time. Engaging in outdoor play is correlated with lower levels of anxiety and improved behavioral outcomes. This finding is consistent with the literature that advocates for the mental health benefits of physical activity, including enhanced mood and reduced stress levels (*Carter et al., 2021*; *Dale et al., 2019*; *Romero-Pérez et al., 2020*). We observe a similar protective effect on academic performance, where higher academic performance negatively correlates with the SAS, SPS, and externalizing problems from the Pediatric Symptom Checklist (EP-PSC), suggesting that better academic outcomes are associated with lower levels of anxiety and behavioral difficulties. These findings underscore the importance of balancing screen time with outdoor activities and academic engagement to support healthier psychological outcomes in children. Collectively, these findings advocate for multifaceted interventions that focus on reducing BMI, increasing physical activity, increasing academic performance, and managing screen time to address and improve children's behavioral health effectively. Screen time is linked to BMI and poor diets, especially in Mediterranean countries where screen time is linked to reduced fruit and vegetable intake (*Wärnberg et al., 2021*).

Surprisingly, in this study age and screen time had no relationship which is contrary to some studies reporting a potential link between age and biological maturity during adolescence with increased screen time (*Kerai et al., 2022*; *Nagata et al., 2023*). However, this study did not record biological maturity or factors like puberty-related boredom or social pressure which could be possible confounders. Further, the association between the dependent variable (screen time) and independent variables was examined using linear regression models. Hence, the readers shall execute caution while interpreting the findings based on the limitations of the model like multicollinearity, assumption of linearity, and assumptions of homoscedasticity.

## Limitations and strength

The mention of few limitations of this study would allow the readers to execute caution while interpreting the findings. First, other confounding factors could affect a child's academic performance at school. Individual factors such as intellectual ability, learning styles, motivation, and overall health significantly impact students' learning outcomes and parent education. Second, parental involvement & supervision, the disparity between recreational screen time *vs.* the one for education, socioeconomic status, family structure, the educational attainment of parents, and the resources available to children are the family factors that were not considered in this study (*Farooq et al., 2011*;

*Habibullah & Ashraf, 2013*; *Pinquart & Ebeling, 2020*). Finally, the cross-sectional design of this study did not allow exploration of causal-effect relationships between variables, proxy-reported measures of screen time, and confounders like socioeconomic status, family type, and siblings were not studied. Nevertheless, this study explored the associations of the screen time of children with their mental health and academic performance. Future research shall focus on parental involvement, parental screen time, and the disparity of SES associated with screen time. Exploration of screen time among children using objective outcome measures should strengthen the evidence of associations further. The findings of this study can be generalized to countries with similar digital educational systems and the readers shall execute caution while interpreting the results based on the limitations mentioned.

## Strategies reported in the literature for managing screen time

Excess screen time is associated with increased childhood obesity and negatively affects behavior. Addressing the challenges posed by excessive screen time among children necessitates a collaborative approach involving educational systems, parents, and public health initiatives.

On the basis of our findings and the key strategies identified from the literature during this study, we believe that the findings will surely benefit stakeholders in the management of this key problem.

*Education policy:* develop curricula that use technology to enhance learning rather than replace traditional methods. This includes the use of interactive tools that promote engagement and understanding (*Calderón, Merono & MacPhail, 2020*). *Train educators:* teachers are equipped with the skills to use digital tools effectively in the classroom, ensuring that they can guide students in balanced technology use (*Rodrigues, 2020*). *Scheduled tech-free times:* implement designated periods during the school day that are free from digital devices to encourage face-to-face interaction and physical activity (*Sharma & Sharma, 2024*). *Establishing clear screen time rules:* specific daily or weekly screen time limits for children should be set by parents, especially to reduce screen use before bedtime to help improve sleep quality. *Encourage engaging in alternatives:* promotion activities that can replace screen time, such as reading, sports, and the arts, to provide diverse experiences that support cognitive and physical development. *Be a role model:* parents should also moderate their screen use to set a behavioral standard, showing children that there is a time and place for technology. *Awareness campaigns:* launch educational campaigns that inform parents and children about the risks of excessive screen use, such as increased anxiety and behavioral issues, as well as the benefits of alternative activities. *Community programs:* support or introduce community initiatives that provide children with opportunities for outdoor play and physical activities, which have been shown to reduce anxiety and improve academic performance. *Screen time recommendations:* publicize guidelines on the optimal screen time for different age groups to help parents and educators make informed decisions (*Whiting et al., 2021*). *Observation and feedback:* regular screen time audits encourage families to regularly review and discuss screen time habits and make adjustments as needed. Feedback Mechanisms in Schools: schools can

offer feedback to students and parents about screen time usage and its impacts, helping them understand the consequences of their screen habits. *Legislation and policy enhancement:* regulate content: work toward stricter regulations on addictive content for children, ensuring that digital content is age-appropriate and educational. Supportive Legislation: Advocates for policies that encourage reduced screen time in educational institutions and promote physical activity in school curricula (*Woods et al., 2021*).

## CONCLUSION

In summary, this study contributes important insights into the impact of screen time on child development, emphasizing the need for careful management of digital engagements to safeguard and promote children's mental and academic growth. As technology continues to pervade every aspect of daily life, ongoing research and adaptive interventions will be essential to ensure that digital media serves as a positive force in children's educational and developmental trajectories.

## ACKNOWLEDGEMENTS

We acknowledge the Galgotias Multi-Disciplinary Research and Development Cell (G-MRDC) for the facility support and our deepest gratitude to the school authorities and parents who participated in this study.

### Funding

This study was supported by the Prince Sattam bin Abdulaziz University project number (PSAU/2025/R/1446), Saudi Arabia. The funders had no role in study design, data collection and analysis, decision to publish, or preparation of the manuscript.

### Grant Disclosures

The following grant information was disclosed by the authors:
Prince Sattam bin Abdulaziz University, Saudi Arabia: PSAU/2025/R/1446.

### Competing Interests

Faizan Kashoo is an Academic Editor for PeerJ.

### Author Contributions

- Mohammad Sidiq conceived and designed the experiments, performed the experiments, prepared figures and/or tables, and approved the final draft.
- Balamurugan Janakiraman conceived and designed the experiments, analyzed the data, prepared figures and/or tables, authored or reviewed drafts of the article, and approved the final draft.
- Faizan Kashoo conceived and designed the experiments, analyzed the data, prepared figures and/or tables, authored or reviewed drafts of the article, and approved the final draft.

- Rayan Jastania conceived and designed the experiments, authored or reviewed drafts of the article, and approved the final draft.
- Abdullah Ibrahim Alhusayni analyzed the data, authored or reviewed drafts of the article, and approved the final draft.
- Abdullah Alzahrani conceived and designed the experiments, authored or reviewed drafts of the article, and approved the final draft.
- Aksh Chahal performed the experiments, analyzed the data, prepared figures and/or tables, and approved the final draft.
- Alagappan Thiyagarajan conceived and designed the experiments, authored or reviewed drafts of the article, and approved the final draft.
- Imran Khan analyzed the data, authored or reviewed drafts of the article, and approved the final draft.
- Chandan Kumar performed the experiments, prepared figures and/or tables, and approved the final draft.
- Rajkumar Krishnan Vasanthi conceived and designed the experiments, performed the experiments, authored or reviewed drafts of the article, and approved the final draft.
- Fahad Alanazi conceived and designed the experiments, analyzed the data, prepared figures and/or tables, and approved the final draft.
- Mehrunnisha Ahmad performed the experiments, analyzed the data, prepared figures and/or tables, and approved the final draft.
- Chhavi Arora Sehgal conceived and designed the experiments, performed the experiments, authored or reviewed drafts of the article, and approved the final draft.
- Shabnam Khan conceived and designed the experiments, performed the experiments, authored or reviewed drafts of the article, and approved the final draft.
- Mshari Alghadier conceived and designed the experiments, prepared figures and/or tables, authored or reviewed drafts of the article, and approved the final draft.

### Human Ethics

The following information was supplied relating to ethical approvals (*i.e.*, approving body and any reference numbers):

Departmental Research Committee, School of Allied Health Sciences, Galgotias University, under reference to DEC/FEA/PT/06/23, and the study was registered with the Indian Council of Medical Research India (CTRI/2023/10/058965).

### Data Availability

The raw data is available in the Supplemental File.

### Supplemental Information

Supplemental information for this article can be found online at http://dx.doi.org/10.7717/peerj.19409#supplemental-information.

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
