# Peer review of "Screen time exposure and academic performance, anxiety, and behavioral problems among school children"

_PeerJ, doi:10.7717/peerj.19409_

## Round 0.1 · original submission · Minor Revisions

Dear Authors,
I am pleased to forward the reviewer's comments on your manuscript. Their insights offer valuable perspectives for enhancing your work. We look forward to receiving your revised manuscript, addressing these suggestions, at your earliest convenience.

Reviewer 1 ·

Basic reporting

COMMENTS TO THE AUTHOR

Comment 1. The article seeks to investigate the relationships between screen time and academic performance, anxiety, and outdoor playing among school children in India. This is an interesting question and it is vital that more good quality research is published on the health and lifestyle of children, particularly on less studied countries.

Comment 2. The abstract summarizes the paper properly. However, some clarifications are required in the methods section and in its main conclusion.

Comment 3: At the introduction section, the reason for this focus was well developed in that paper. However, in my opinion, the authors should add some important rates and results of the different studies; furthermore, a better contextualization of data of the present study as well as better connection between sub-issues of that content. Finally, the hypotheses of the study might be also presented.

Comment 4: At the methods section, there is no mention or consideration for differences in the timing of biological maturity between individuals and groups. Biological maturity is related to physical fitness, BMI and likely physical activity and thus differences in biological maturity (pubertal development) between groups could be confounding results. I would suggest that if the biological maturity information (e.g. self report secondary sex characteristics, predicted years from PHV, predicted percentage of adult height attained) is available that the authors control for this in the analyses.

Comment 5: Furthermore (methods section), the authors have summarized the procedures. However, there are some details which could be added; for example, the validity and reliability of the tools/questionnaire used [i.e. The Spencer Children Anxiety Scale (SCAS-Child)].

Comment 6: The results are presented clearly, but please, avoid redundant content with the tables.

Specific Comment 7: The discussion section could be improved since in some parts appear to simply be a confirmation of the results; there is too much focus upon describing the results rather than discussion/ interpretation/ explanation of findings.

Specific comment 8: I would like to see the authors discuss in detail other sources of variation, particularly related to the socio and educational variables, and so on!

Specific Comment 9: The conclusion is presented in a pragmatic way.

Specific Comment 10: In general, the quality of writing is satisfactory, but in some places of the manuscript could be improved.

Specific Comment 11: The references are ok.

Experimental design

.

Validity of the findings

.

Reviewer 2 ·

Basic reporting

The study examines the effects of screen time on academic performance, anxiety, and behavior in school-aged children. While the general structure is solid, there is room for improvement.

-The title adequately reflects the content of the study.

-The tables presented (Table 1, Table 2, Table 3) are informative; however, it is recommended to improve their footnotes to allow for better independent comprehension.

-The relationship between screen time and BMI appears to be associated also by inadequate eating habits - a phenomenon reported in Mediterranean contexts where higher screen exposure is associated with lower fruit and vegetable consumption. I suggest this could be mentioned both in the introduction as in the discussion. The following reference could be included to support this observation: https://doi.org/10.3390/jcm10040795. This reference could also be added to lines 82, 216, and 251 to further substantiate the cited evidence.

Experimental design

The methodology employed is generally appropriate; however, further clarifications and justifications are required in certain aspects.

Inclusion criteria: The inclusion criteria are briefly mentioned (children aged 8–14 years from five schools), but it is recommended to provide more details regarding the school selection process. For example: How many schools were invited? Were public schools invited, or only private ones? How many schools declined participation? These details are essential to determine whether the sample is representative or specific to a particular population. Describe the caracteristics of the 5 included schools.

Sample size: The sample size (n=537) appears adequate; however, it would be beneficial to include a statistical justification for this sample size.

Instruments: The description of the instruments used (SCAS and PSC) is appropriate, but if there is some additional information on their previous use within the Indian context, this should be provided.

Data collection procedure: The data collection method via an online questionnaire is well-described; however, it is recommended to discuss potential limitations associated with this approach. Parental reporting vs child self report?

A clear limitation and confounding factor in this study is the parental education and involvement. This is already mentioned in the abstract but requires improvement in lines 267–270. Additionally, in line 235, the authors claim that this is a topic for future investigation; however, it should be noted that this has already been demonstrated (see reference: https://doi.org/10.3390/jcm10040795).

Study Limitations: The limitations of the study are briefly mentioned at the end but would benefit from a more detailed discussion.

Validity of the findings

The findings presented are generally valid; however, certain aspects require deeper discussion.

Regression Analysis: The linear regression analysis (Table 3) supports the conclusions regarding the relationship between screen time and outcome variables. Nevertheless, it is recommended to discuss the limitations of this analysis and address the absence of confounding variables.

Confounding Variables: It is suggested that potential confounding variables—such as socioeconomic status or parental education level—be addressed more thoroughly in the discussion.

---

## Round 0.2 · Minor Revisions

Thank you for your revised manuscript and responses to the reviewers.

Following the reviewer's comment regarding the sample size justification, I kindly ask you to include the requested statistical analysis in the Methods section.

Please let us know if you have any questions, and I look forward to receiving your updated manuscript.

Reviewer 2 ·

Basic reporting

Dear authors, thank you for your response.

Experimental design

While I understand the practical approach for widespread survey distribution, the absence of a statistical justification for the sample size remains a limitation. To meet methodological standards, please provide either: (1) an a priori power calculation showing the target sample size needed to detect clinically meaningful effects (even if retrospectively applied to your design), or (2) a post hoc sensitivity analysis demonstrating the minimum detectable effect sizes (e.g., correlations or ORs) achievable with your final sample (n=537) at 80% power and α=0.05. This is critical to confirm your study was adequately powered for its primary outcomes. I strongly recommend adding this analysis to the Methods section to bolster the validity of your conclusions.

Validity of the findings

See comments mentioned above.

---

## Round 0.3 · accepted · Accept

Thank you for your corrections. Congratulations!